# Genetic Variability of the Internal Transcribed Spacer and Pyruvate:Ferredoxin Oxidoreductase Partial Gene of *Trichomonas vaginalis* from Female Patients

**DOI:** 10.3390/microorganisms11092240

**Published:** 2023-09-05

**Authors:** Fernando Martinez-Hernandez, Fabiola Sanchez-Aguillon, Joel Martinez-Ocaña, Nelly Raquel Gonzalez-Arenas, Mirza Romero-Valdovinos, Eduardo Lopez-Escamilla, Pablo Maravilla, Guiehdani Villalobos

**Affiliations:** 1Departamento de Ecologia de Agentes Patogenos, Hospital General “Dr. Manuel Gea Gonzalez”, Mexico City 14080, Mexico; fherxyz@yahoo.com (F.M.-H.); joelmoc_7037@yahoo.com.mx (J.M.-O.); nelly_raquel@hotmail.com (N.R.G.-A.); eduar1escamilla@gmail.com (E.L.-E.); 2Laboratorio de Investigación del Departamento de Biologia Molecular e Histocompatibilidad, Hospital General “Dr. Manuel Gea Gonzalez”, Mexico City 14080, Mexico; ofalabi_j@hotmail.com; 3Laboratorio de Patogenos Emergentes, Departamento de Biologia Molecular e Histocompatibilidad, Hospital General “Dr. Manuel Gea Gonzalez”, Mexico City 14080, Mexico; mirzagrv@yahoo.com; 4Laboratorio de Biologia Molecular del Departamento de Produccion Agricola y Animal, Universidad Autonoma Metropolitana, Mexico City 04960, Mexico

**Keywords:** *Trichomonas vaginalis*, pyruvate:ferredoxin oxidoreductase (PFOR), internal transcribed spacer (ITS)

## Abstract

In the present study, we evaluated the genetic variability of the internal transcribed spacer (ITS) region and the pyruvate:ferredoxin oxidoreductase (*pfor*) A gene of *Trichomonas vaginalis* from female patients and its possible implications in the host–parasite relationship. Phylogenetic and genetics of populations analyses were performed by analyzing sequences of the ITS region and partial *pfor* A gene of clinical samples with *T. vaginalis*, as previously documented. Alignments of protein sequences and prediction of three-dimensional structure were also performed. Although no correlation between the main clinical characteristics of the samples and the results of phylogeny was found, a median-joining analysis of ITS haplotypes showed two main clusters. Also, *pfor* A, due to its phylogenetic divergence, could be used as a marker to confirm the genus and species of trichomonads. Alignment of protein sequences and prediction of three-dimensional structure showed that PFOR A had a highly conserved structure with two synonymous mutations in the PFOR domain, substituting a V for a G or a S for a P. Our results suggest that the role of genetic variability of PFOR and ITS may not be significant in the symptomatology of this pathogen; however, their utility as genus and species markers in trichomonads is promising.

## 1. Introduction

Trichomoniasis by *Trichomonas vaginalis* is the most common nonviral sexually transmitted infection (STI) in humans worldwide; in 2016, the WHO estimated 156 million new infections [1]. *T. vaginalis* infection in females is symptomatic in approximately 50% of cases, and approximately 30% of patients with asymptomatic cases develop some symptoms in the 6-month period postinfection [2]. A wide variety of clinical symptoms have been described in women, such as urethral discharge, itching, abdominal pain and dysuria, which can lead to premature birth and low birth weight in newborns. In contrast, infection of the male genitourinary tract is generally asymptomatic, although mild urethritis, epididymitis and prostatitis can occur [3,4].

Under both aerobic and anaerobic conditions, *T. vaginalis* uses carbohydrates as its main energy source via fermentative metabolism. Carbohydrate metabolism occurs in the cytoplasm and hydrogenosome, which is analogous to mitochondria and is the site of fermentative oxidation of pyruvate [4]. Anaerobic energy metabolism is accomplished through catalyzed reactions by several key enzymes, such as pyruvate:ferredoxin oxidoreductase (PFOR) [5]. In addition, the draft of the *T. vaginalis* genome sequence revealed *pfor*-like genes such as *pfor* A, *pfor* BI, *pfor* BII and *pfor* E [6]; these genes are available in the recent released *T. vaginalis* genome sequence (GenBank project number PRJNA885811). In different organisms, anaerobic energy enzymes are known to function in different subcellular compartments in such a way that they perform divergent functions as a consequence of their localization and environmental conditions [7,8], as observed for PFOR in *T. vaginalis* [9], which is located on the membranes of hydrogenosomes and is responsible for the decarboxylation of pyruvate to acetyl-CoA [10]. Although the catalytic mechanism of PFOR is not fully understood, it is assumed that Fe-S groups are involved in the catalytic mechanism of the enzyme. In bacteria, preserved motifs of Fe-S centers with cysteines (Cys), which coordinate the Fe-S centers present in the enzyme, have been described [5]. In addition, this enzyme is an example of a surface-associated cell-binding protein that has enzyme activity and is involved in cytoadherence [11]. Parasite adherence to vaginal epithelial cells is fundamental for initiating and maintaining *Trichomonas* infection and for survival in the host [12]. PFOR is involved in proliferation, adhesion to host cells and abscess formation, so it could influence the pathogenicity of this parasite [9,12,13]. Interestingly, this enzyme is present in protozoan parasites but is absent in humans [14]. It is important to highlight that, to date, PFOR studies have been focused on their characterization and function depending on their localization, but whether *pfor* genes present genetic variations and are associated with pathogenicity or the development of specific symptoms have not been described.

On the other hand, the internal transcribed spacers (ITSs), ITS1 and ITS2, are spacer regions divided by the 5.8S rRNA gene (ITS1+5.8S+ITS2); these loci evolve much faster than coding regions because substitutions occurring in spacers may be considered neutral mutations without any constraints. Thus, there is a large amount of information related to the usefulness of ITS, mainly clarifying problematic taxa, such as cryptic species [15]. Although there are some reports about the usefulness of ITS1-5.8S rRNA-ITS2 as a marker for differentiation among *T. vaginalis* isolates, information obtained using this locus is still scarce [16,17,18,19]. Therefore, the objective of the present study was to evaluate the genetic variability of the ITS1+5.8S+ITS2 region and the pyruvate:ferredoxin oxidoreductase (*pfor*) A gene of *T. vaginalis* from female patients and its possible implications in the host–parasite relationship.

## 2. Materials and Methods

### 2.1. Ethical Approval and Sampling

The Ethics and Research Committees at Hospital General “Dr. Manuel Gea Gonzalez” (HGMGG) approved this study, with reference number 12-75-2015. Recovered samples, used primarily for pathogen diagnosis of 28 women with trichomoniasis who attended the gynecology service of the HGMGG between 2015 and 2019, were analyzed [20]; thus, vaginal swab samples confirmed by microscopy with *T. vaginalis* were placed in sterile plastic tubes with 0.9% physiological saline solution and kept at −70 °C until DNA isolation.

### 2.2. PCR Sequencing

DNA was extracted from approximately 500 µL of the physiological saline solution in which vaginal swabs were immersed using the Puregene DNA Purification System (Gentra Systems, Hilden, Germany) according to the manufacturer’s protocol. In samples with unclear microscopic observations, the presence of *T. vaginalis* was confirmed by analyzing the ITS1+5.8S+ITS2 region using primers and PCR conditions that were previously described [21].

To analyze the *pfor* gene, specific primers for variant A of different *T. vaginalis* sequences were designed at the laboratory based on information available in GenBank (sequences AY661465, U16822 and XM_001582310). A suitable region of ~852 bp was chosen for amplification with the forward primer 5′ TCAAGGTCCACCTCTTCCGCCC 3′ and the reverse primer 5′ GCATGGTCGACCATGTCCCAG 3′. This region includes the important sites of the enzyme, such as the domain II and III of the ferredoxin oxidoreductase site, which are all located between amino acids 305 and 603 of the protein. PCR amplifications were carried out in a final volume of 25 μL containing each primer at 10 pmol, 1X PCR buffer (8 mM Tris-HCl, pH 8; 20 mM KCl), 1.5 mM MgCl_2_, 0.5 mM dNTPs and 2 U of Taq DNA Polymerase (Promega, Madison, WI, USA). Approximately 500 ng of DNA was used as a template to amplify genomic sequences. The amplification conditions were 1 cycle at 94 °C for 5 min; 35 cycles including denaturation, annealing and extension steps at 94 °C for 30 s, 54 °C for 1 min and 72 °C for 30 s, respectively; and a final extension step at 72 °C for 7 min. The presence of amplicons was observed by staining with ethidium bromide followed by electrophoresis in a 1.5% agarose gel; in most samples, amplification assays were performed at least twice, both forward and reverse, to achieve the quality and concentration required for sequencing, then the amplicons were purified using an AxyPrep PCR clean-up kit (Axigen Biosciences, Union City, CA, USA) and sequenced on both strands in an ABI prism sequencer (Applied Biosystems, Foster City, CA, USA) using the Sanger method, and chromatograms were obtained.

### 2.3. Phylogenetic and Genetic Variation Analysis

The chromatograms for each sequence were evaluated with the Mesquite V2.5 software using the Chromaseq package [22] that partially makes use of the Phred and Phrap algorithms for base calling, assigning quality values to each called base and assembling contigs [23], because more than one sequence was obtained for most of the samples, and from them obtains a consensus. All sequences were subjected to a BLAST search of the GenBank database and submitted with the access numbers OR493214, OR069659-OR069683 for *pfor* A partial sequences and OR005496-OR005511 for the ITS1-5.8S rRNA-ITS2 region; multiple alignments were performed using CLUSTAL W and MUSCLE programs with manual adjustment in the MEGA program [24]. The molecular evolution model used to build the phylogenetic tree was the Hasegawa–Kishino–Yano model with a gamma distribution and invariant sites (HKY+G+I) for ITS sequences and the general time-reversible model with a gamma distribution and invariant sites (GTR+G+I) for the *pfor* A partial gene, which were selected using MODELTEST software version 3.7 [25]. The algorithms used to carry out the phylogenetic analyses were maximum likelihood (ML), maximum parsimony (MP) and the Bayesian algorithm (BA). ML and MP were performed with 1000 bootstrap replicates under the Tamura–Nei model of evolution, and the algorithms were implemented in MEGA7. BA reconstructions were performed using the MrBayes software version 3.2.7 [26], while the analysis was performed for two million generations with sampling trees every 100 generations including two independent assays with four strands each, considering codon splits. Trees with scores below the burn-in phase were discarded, and those remainders were collected and used to build majority consensus trees. A median-joining network analysis was performed using NETWORK 4.611 with sequences of *T. vaginalis* downloaded from GenBank, which showed high identity with the samples obtained in the present study according to the BLAST search. Haplotype networks were established under default settings and assumptions; sequences of the ITS1+5.8S+ITS2 region sequences of the ITS1+5.8S+ITS2 region for BA reconstructions were MK172845-7, KM095107; KX459478-79, 86; KX977518-20 and KC215389-90 from *T. gallinae*, *T. tenax*, *T. brixi* and *T. stableri*, respectively, using *T. stableri* as the outgroup, while for the *pfor* A tree, sequences of *T. tenax* and *T. gallinae* (OCTD01001480 and MRSU01008911, respectively) were included as the outgroup. In addition, some *T. vaginalis* sequences were downloaded from GenBank as controls.

A genetic diversity analysis among populations was performed using DnaSPv4 [27]; specifically, expected heterozygosity (Hd), nucleotide diversity (π), haplotype polymorphism (θ), Tajima’s D and Fu and Li’s D were calculated. The meaning of these keys are as follows: π, average number of nucleotide differences among all possible pairs of sequences in the sample; θ, proportion of nucleotide sites that can be predicted to be polymorphic from this region of the genome—both indices are used to measure polymorphisms ranging from 0 to 1. Finally, negative values for Tajima’s D and Fu and Li’s D tests allow us to test for past population expansion or neutrality of mutations; to this end, negative values suggest a recent expansion process or an effect of purifying selection [28,29].

To search for an association between the phylogenetic analysis of both markers (ITS and *pfor* A) and the main clinical characteristics of the samples, since the small sample size and frequency in at least one cell was less than 5, Fisher’s exact tests for two tails were performed, and odds ratios and 95% confidence intervals were also calculated using Epi-Info6 software v6.04 (Centers for Disease Control and Prevention, Atlanta, GA, USA).

### 2.4. Alignment of Protein Sequences and Prediction of Three-Dimensional Structure

The sequences obtained with the primers specific to the A variant of the *pfor* gene were translated to amino acid sequences and aligned with those for PFOR of *T. vaginalis* Q4KY23, Q27088 and A0A8U0WP41 deposited in the UniProt database. This analysis was performed using the ClustalW program. The I-TASSER [30] server was used to model the 3D structure of *T. vaginalis* PFOR variant A. This server is under active development with the intention of providing the most accurate predictions of protein structure and function using state-of-the-art algorithms. After analysis, the models with the highest confidence scores (C-scores) were selected. Three-dimensional structures were analyzed and visualized using PyMOL 2.5.2 [31].

## 3. Results

### 3.1. Characteristics of the Study Population

Twenty-eight vaginal swab samples of women with a mean age of 34 ± 13.7 years were analyzed; the main gynecological and microbiological findings for the women were previously reported [20]. Briefly, most of the patients described genital burning/itching (35.7%), 21.4% had cervicovaginitis and 17.8% exhibited vaginal fetid discharge and 42.8% were pregnant. Interestingly, in six samples (21.4%), coinfections with *Blastocystis* ST1-3 were found. DNA extractions and PCRs were performed for each sample from two to four times to reach the concentration and quality recommended for sequencing. Unfortunately, since the DNA obtained from the samples was used for a previous study [20], it was not possible to obtain amplicons of ITS1+5.8S+ITS2 and *pfor* of sufficient quality and concentration for sequencing in all cases. All sequences were subjected to a BLAST search against the GenBank database (OR493214, OR069659-OR069683 to *pfor* A partial sequences and OR005496-OR005511 for the ITS1-5.8S rRNA-ITS2 region).

### 3.2. Phylogenetic and Genetic Variation Findings

Sixteen samples sequenced for the ITS1+5.8S+ITS2 region showed two haplotypes; the median-joining network tree for this locus (Figure 1) included the haplotypes obtained here as a main cluster (subgroup 1), near the divergent cluster of *T. tenax*. Interestingly, a second cluster of *T. vaginalis* haplotypes was identified (subgroup 2), close to the *T. gallinae* cluster.

The phylogenetic tree grouped all samples analyzed with the *T. vaginalis* clade with strong support (97/99/1.00); although there were no strong support values, three subclades were also observed. In contrast, other trichomonads, such as *T. gallinae*, *T. tenax*, *T. stableri* and *T*. *brixi,* were clearly separated (Figure 2).

Sequences for a partial *pfor* A gene of 26 samples were obtained, and some haplotypes were identified; no correlations between the main clinical characteristics of the samples and the results of phylogenetic analysis for *pfor* A were found (Appendix A, Table A1). The median-joining network tree for the partial *pfor* A gene (Figure 3) showed a wide dispersion of haplotypes, without specific clustering. All *pfor* A haplotypes of *T. vaginalis* samples were included in a specific clade (100/100/1.00), separated from *T. gallinae* and *T. tenax*; although three small subclades were distinguished, the bootstrap support values were below 70 (Figure 4).

Genetic polymorphism indices of both studied genes were calculated, and Table 1 summarizes the results. The expected heterozygosity values for both markers (ITS or *pfor* A) ranged from 0.49 to 0.79; in contrast, for the π and θ indices, very low values (~0.002) were obtained. Regarding Tajima’s D and Fu and Li’s D, most of the values were negative but without statistical significance, except for those of the ITS1+5.8S+ITS2 haplotypes, which were statistically significant. Interestingly, when ITS subgroups were independently analyzed, subgroup 2 showed higher variability (π > 0.005) than subgroup 1; although Tajima’s D and Fu and Li’s D tests showed negative values for both subgroups, only subgroup 2 values had significant support.

### 3.3. Alignment of pfor A and Prediction of Its Three-Dimensional Structure

Regarding the protein deduction analysis, the nucleotide sequences of *pfor* A were translated, yielding sequences of approximately 284 amino acids. Protein alignments of such sequences showed four highly conserved cysteine residues. In addition, two mutations were found, one in the OR069675 sample of Val by Gly-137 and another in Q4KY23 of Ser by Pro-463 (Appendix B, Figure A1). The predicted three-dimensional structure of the protein (Figure 5A) shows the domain of pyruvate ferredoxin oxidoreductase in cyan color, consisting of six β-strand structures, exhibiting mutations in red and four α-helix structures. The consensus sequence Val and Ser (Figure 5B), and mutations to Gly and Pro (Figure 5C,D, respectively) can also be observed.

## 4. Discussion

Several factors have been documented that can modify the virulence of *T. vaginalis*, such as transcriptomics and proteomics of the adhesion process [32], the presence of the double-stranded RNA virus known as *T. vaginalis* virus (TVV) [33], interactions with human-associated bacteria such as *Mycoplasma hominis* [34] infection and the promotion of vaginal microbiota imbalance by increasing the proportions of *Parvimonas, Sneathia* and other anaerobes [35]; however, association among the symptomatology and the genetic variability of ITS and *pfor* A gene of *T. vaginalis* clinical isolates has not been documented.

PFOR is a metabolic enzyme described in several anaerobic microbial eukaryotes, such as *Entamoeba histolytica* [36], *Giardia duodenalis* [37] and *T. vaginalis* [9,38]. In trichomonads, it has been highlighted for possessing alternative, nonenzymatic functions [39]; thus, PFOR is also associated with the process of pathogenicity in *T. vaginalis*, such as adherence to host cells and proliferation [9,39]. In the present study, no correlations between the main clinical characteristics of the samples and the results of phylogenetic analysis for *pfor* A were found. As is often the case in many studies, phylogenetic inferences do not show clear agreement between the evolution of the pathogen and the development of the disease [40].

Although in the present study the ITS1+5.8S+ITS2 loci were originally used as a tool for the molecular identification of *T. vaginalis*, the sequences obtained and their subsequent analysis provided interesting information about the population genetic structure of this pathogen. The phylogenetic tree grouped our samples within the *T. vaginalis* main cluster; although subclustering for the main cluster were observed, the low bootstrap value (below 70) suggest that this subclustering should be interpreted with caution; in contrast, the median-joining network allowed us to sort all sequences into two subgroups: one with less variability and from which *T. gallinae* diverged and another undergoing an expansion event and that is close to the *T. tenax* cluster. On the other hand, an interesting global population genetics study using a panel of microsatellite markers and SNPs comprising isolates from different parts of the world, including 11 samples from Mexico, was performed [33]. In that study, it was shown that *T. vaginalis* is a genetically diverse parasite with a unique population structure, exhibiting two types (type 1 and type 2) present in equal proportions worldwide; therefore, our results for the ITS1+5.8S+ITS2 loci are consistent with those presented by Conrad et al. [33].

For *pfor* A sequences of *T. vaginalis*, a robust phylogenetic tree was generated that clustered all sequences in a clear and specific cluster, away from the *T. gallinae* and *T. tenax* clusters; thus, our results suggest that this gene could eventually be used as a marker to confirm genus and species identities among trichomonads. However, because there are few sequences available for this gene in the *Trichomonas* genus, further studies are required to confirm this finding. When two different loci are submitted to a phylogenetic analysis, it is common to observe dissimilarities through inference [40]. A study focused on the genetic variability of *pfor* and the small-subunit ribosomal RNA (18S rRNA) from *Blastocystis* sp. also found similarities between the phylogenies of the two markers [41]. In another study, during the sequencing analysis of the Fe-hydrogenase gene of *T. gallinae* isolates, the median joining network analysis showed a distribution of haplotypes similar to the median joining network inference for *pfor* halplotypes of *T. vaginalis* obtained in the present study; both Fe-hydrogenase and PFOR are enzymes placed in the hydrogenosomes of *Trichomonas* spp. [42]

Regarding the genetic variability of ITS and *pfor* A, both sequences showed very low values (~0.002) for the π and θ indices. In addition, Tajima’s D and Fu and Li’s D tests showed negative values with statistical significance only for ITS1+5.8S+ITS2 haplotypes. A study on the identification of genetic variants of *T. vaginalis* in China, analyzing fragments of the 18S rRNA, found that π values ranged from 0.002 to 0.005 among different isolates, with a significant negative value for Fu’s test [29]. Thus, our results are in concordance with those of Mao and Liu [43] and with those of Conrad et al. [33] because they support the finding that, as noted above, *T. vaginalis* is a genetically diverse parasite with a unique population structure, exhibiting two types (types 1 and 2), while the Mexican samples were grouped exclusively into a single type, which could be undergoing a process of expansion.

On the other hand, since the diversity and phylogeny of *pfor* A may allow us to infer the function of the protein or may be related to characteristics of the population analyzed (i.e., in *Blastocystis* sp., *pfor* did not allow discrimination between subtypes of parasites), the phylogeny of this enzyme may provide information about the biochemical function instead of the relationship of the group [41].

The alignment of the amino acid sequences of the samples obtained in this study with the PFOR sequences of *T. vaginalis* Q4KY23, Q27088 and A0A8U0WP41 showed four highly conserved cysteines; interestingly, these cysteines are far from each other, i.e., CX90-CX60-CX87-CX, and there is no similarity to any cysteine-rich motif reported in the literature. Since our results showed that Gly and Pro mutations in the PFOR sequence are not located near the cysteine-rich sites mentioned above, it is likely that these mutations have no impact on the PFOR activity of *T. vaginalis*; in addition, some mutations in nitroreductases (enzymes involved in the processing of metronidazole) of *T. vaginalis* have been identified, e.g., mutations from Val to Asp-51, Val to Ala-153, Val to Ala-159, Val to Ala-117, Val to Ala-132 and Ser to Gly-29, and only the Val mutation to Ala-59 has been associated with resistance to metronidazole [44]. Thus, with the design of the present study, it is not possible to clarify the role of the PFOR mutations.

Until now, the heterogeneity of clinical manifestations of trichomoniasis associated with differences in phenotypic and genetic expression of *T. vaginalis* has not been clear [16,17,43]. The population or isolates of *T. vaginalis* show significantly different levels of genetic variation; thus, the proper characterization of these populations is vital to avoid generalizing the behavior of each trichomonad, as observed for the Mexican population. Understanding these relationships is essential for future strategies to control or prevent *T. vaginalis* in each region.

Since the present study was based on sequence analysis of a fragment of the *pfor* A gene and ITS marker from clinical isolates of *T. vaginalis* recovered from 26 patients, the small sample size constitutes a potential bias in the analysis of their genetic variability; however, in both markers it was possible to yield robust phylogenetic inferences that allows identification among different clusters and haplotypes as well as a probable population expansion process in this parasite. In addition, the development of future studies using next-generation sequencing (NGS) for a more extensive search and accurate results applied to a case-control design would help to clarify the etiological role of these markers in clinical isolates of *T. vaginalis* and include the study of samples from other populations and male patients.

## 5. Conclusions

Our results suggest that the role of genetic variability of *pfor* and ITS may not be significant in the symptomatology of this pathogen; however, their utility as genus and species markers in trichomonads is promising.

## Figures and Tables

**Figure 1 microorganisms-11-02240-f001:**
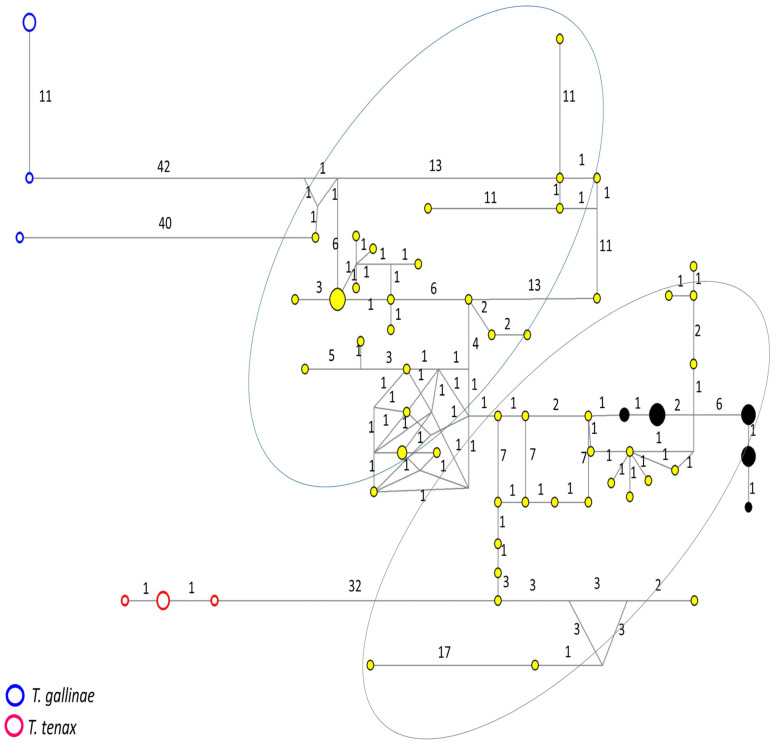
Haplotype network tree generated using ITS1+5.8S+ITS2 region sequences of *T. vaginalis* from Mexico and GenBank; the numbers on branches refer to mutational changes. Haplotypes for *T. gallinae* are shown in blue, for *T. tenax* in red, for Mexican samples obtained here in black and downloaded from GenBank in yellow. Two subgroups (1 and 2, respectively) can be differentiated, one close to the species *T. gallinae* and another to *T. tenax*; both subgroups are shown by two large circles. Numbers in branches are mutational changes; sizes of circles are proportional to haplotype frequencies.

**Figure 2 microorganisms-11-02240-f002:**
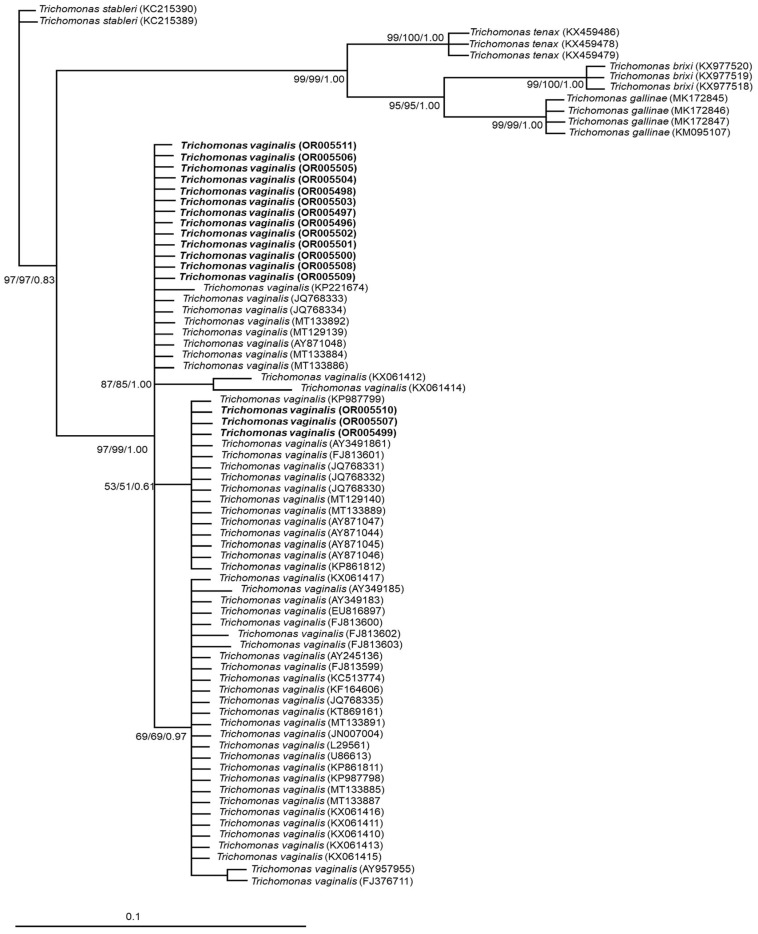
Phylogenetic tree based on ITS1+5.8S+ITS2 region sequences of *T. vaginalis* from Mexico and GenBank. The numbers at the nodes indicate bootstrap support values and Bayesian posterior probabilities, respectively, given by ML, MP and Bayesian analyses. The sequences obtained in this study are shown in bold. Bar = estimated number of substitutions per site.

**Figure 3 microorganisms-11-02240-f003:**
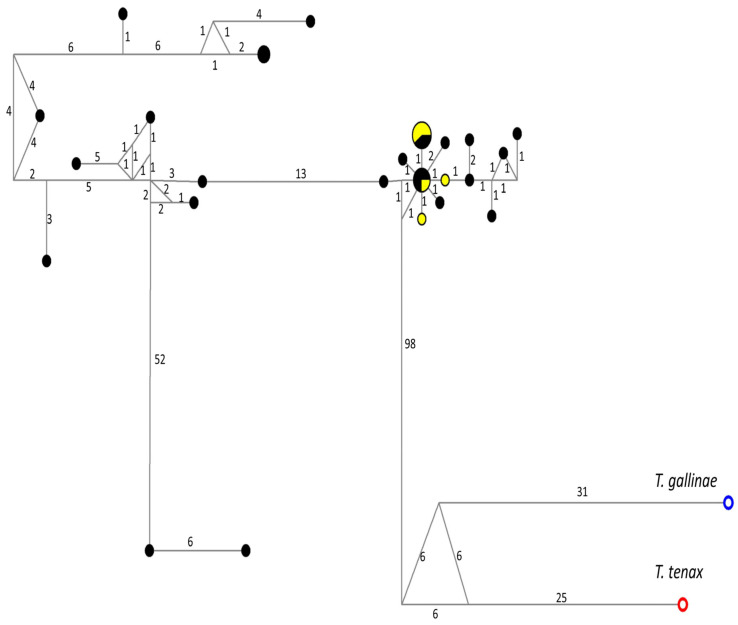
Haplotype network tree generated using *pfor* A sequences of *T. vaginalis* from Mexico and GenBank; the numbers on branches refer to mutational changes. Haplotypes for *T. gallinae* are shown in blue and *T. tenax* in red. Those of Mexican samples obtained here are shown in black, and haplotypes downloaded from GenBank are shown in yellow. Numbers in branches are mutational changes; sizes of circles are proportional to haplotype frequencies.

**Figure 4 microorganisms-11-02240-f004:**
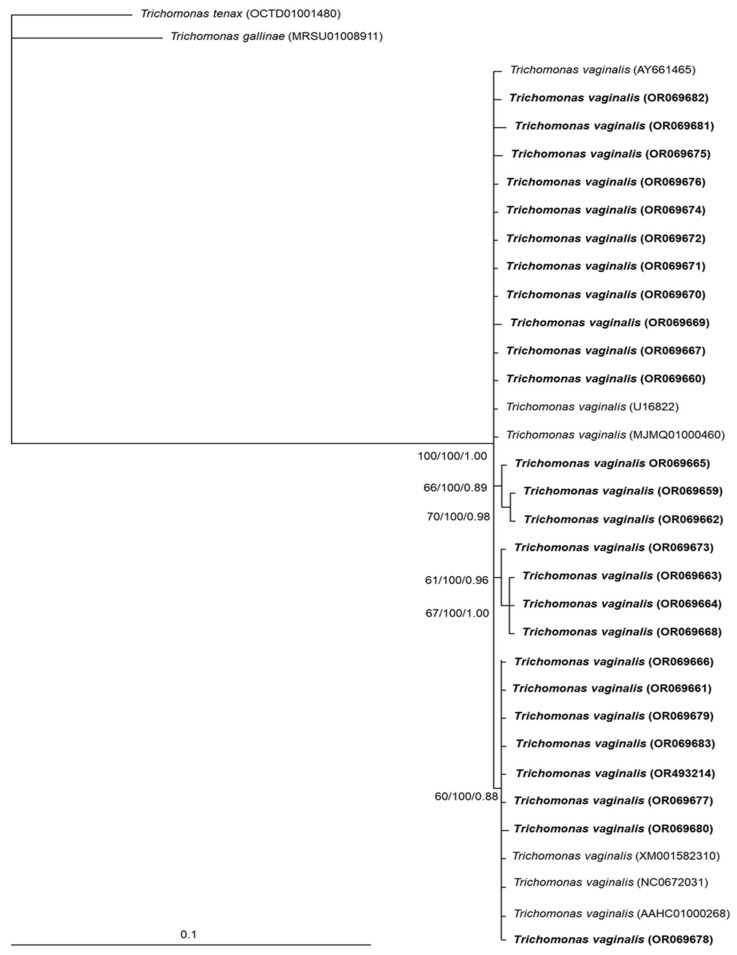
Phylogenetic tree based on *pfor* A sequences of *T. vaginalis* from Mexico and GenBank. The numbers at the nodes indicate bootstrap support values and Bayesian posterior probabilities, respectively, given by ML, MP and Bayesian analyses. The sequences obtained in this study are shown in bold. Bar = estimated number of substitutions per site.

**Figure 5 microorganisms-11-02240-f005:**
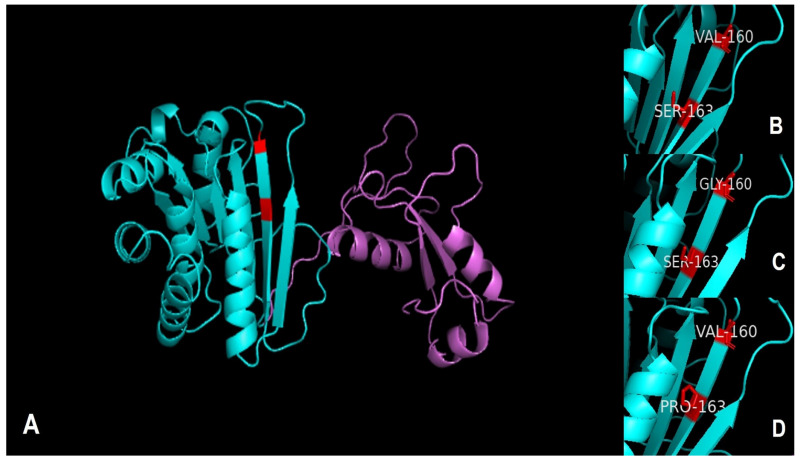
Prediction of three-dimensional structure for PFOR A of *T. vaginalis*. In (**A**), the whole three-dimensional structure is shown. The image shows the domain of pyruvate ferredoxin oxide reductase in cyan. In addition, mutated amino acids are shown in red. Panel (**B**) provides a zoomed-in view of the consensus sequence with Val and Ser. In (**C**), the mutation of Val to Gly is observed, and, in (**D**), the mutation of Ser to Pro is shown.

**Table 1 microorganisms-11-02240-t001:** Genetic polymorphism indices between different *Trichomonas vaginalis* sequences.

Population	n	Expected Heterozygosity (Hd)	Nucleotide Diversity (π)	Haplotype Polymorphism (θ)	Tajima’s D (*p*)	Fu and Li’s D (*p*)
Whole *pfor* A *	32	0.790	0.0019	0.0033	−1.267 (*p* > 0.10)	−1.540 (*p* > 0.10)
Mexican samples *pfor* A	26	0.815	0.0021	0.0031	−1.041 (*p* > 0.10)	−1.069 (*p* > 0.10)
**Whole ITS ‡**	68	0.493	0.0027	0.0087	**−1.942** **(*p* < 0.05)**	**−3.253** **(*p* < 0.05)**
Mexican samples ITS	16	0.325	0.0011	0.0010	0.155 (*p* > 0.10)	0.688 (*p* > 0.10)
ITS type 1	28	0.648	0.0042	0.0087	−1.723 (*p* > 0.10)	−1.997 (*p* > 0.10)
ITS type 2 +	42	0.487	0.0058	0.0022	**−2.443** **(*p* < 0.05)**	**−2.708** **(*p* < 0.05)**

* Five sequences of *pfor* A gene were downloaded from GenBank, and they were added as references. ‡ Fifty-two sequences of the ITS1+5.8S+ITS2 region were downloaded from GenBank, and they were added as references. + Twenty-six sequences of the ITS1+5.8S+ITS2 region were downloaded from GenBank, and they were added as references. Characters in bold indicate statistically significant values

## Data Availability

All relevant data are within the article. The sequences data were submitted to the GenBank database under the accession numbers OR005496-OR005511 for the ITS1-5.8S rRNA-ITS2 region and OR069659-OR069683 to *pfor* locus.

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
