# Peer review of "Genetic Variability of the Internal Transcribed Spacer and Pyruvate:Ferredoxin Oxidoreductase Partial Gene of Trichomonas vaginalis from Female Patients"

_microorganisms, 2023, doi:10.3390/microorganisms11092240_

Round 1
Reviewer 1 Report (Previous Reviewer 1)
The manuscript entitled "Genetic variability of the internal transcribed spacer and pyruvate: ferredoxin oxidoreductase partial gene of Trichomonas vaginalis from female patients" Title, abstract and overall rationale of work is written satisfactory and author fulfill all comments raised by previously. However, there are still some minor concerns, which needs to be addressed before publication.
1) Author added all figures together and that is not attractive. I suggest author should be add after explaining the results for example line no 207 author explain about the figure 1 after that author need to add figure there. Follow same for other figures.
2) What about the figure 5? If author remove from here they should be add in supplementary figure.
3) If figure 5 is not there then author need to correct figure 6 to figure 5 in both text and legend.
English is alright.
Author Response
The manuscript entitled "Genetic variability of the internal transcribed spacer and pyruvate: ferredoxin oxidoreductase partial gene of Trichomonas vaginalis from female patients" Title, abstract and overall rationale of work is written satisfactory and author fulfill all comments raised by previously. However, there are still some minor concerns, which needs to be addressed before publication.
R: Thank you very much for your comments and for your assistance, which has allowed us to improve our manuscript.
1) Author added all figures together and that is not attractive. I suggest author should be add after explaining the results for example line no 207 author explain about the figure 1 after that author need to add figure there. Follow same for other figures.
- Ok, as you can see, in this new version of our manuscript, we inserted figures 1 and 2 after the explanation of ITS trees and figures 3 and 4 after the explanation of pfor A trees
2) What about the figure 5? If author remove from here, they should be adding in supplementary figure.
R: You are right and we apologize, figure 5 has been moved to appendix B, at the end of the manuscript.
3) If figure 5 is not there then author need to correct figure 6 to figure 5 in both text and legend.
R: Thank you, the figures 5 and the Supplementary Figure 1 were corrected, as well as their corresponding figure captions and explanation in the manuscript.
Reviewer 2 Report (Previous Reviewer 2)
After carefully reviewing the manuscript, the authors made several modifications to enhance its overall quality. These changes have resulted in a vastly improved version, which is now deemed suitable for publication without further alterations. The authors have put in a great deal of effort and dedication to ensure that the final product is of the highest standard possible.
Author Response
After carefully reviewing the manuscript, the authors made several modifications to enhance its overall quality. These changes have resulted in a vastly improved version, which is now deemed suitable for publication without further alterations. The authors have put in a great deal of effort and dedication to ensure that the final product is of the highest standard possible.
R: We deeply appreciate your support and feedback. Thank you so much
Reviewer 3 Report (Previous Reviewer 3)
The manuscript by Martinez-Hernandez et al has improved from the previous version, and I would recommend it for publication after major revisions. There are still some inconsistencies that deserve attention.
- First of all, from my previous comments I suggested the authors use the new genome sequence release (PRJNA885811) rather than the previously drafted version (PRJNA16084). My suggestion aims for the authors to provide a more precise gene number and sequence for Pfor, and does not intend, in any case, the authors to sequence the whole genome, or to make any additional/extraordinary financial efforts.
- While the authors have incorporated the isolation source in the title (females), pertinent information has been deleted in the introduction section, where the relevance to women (fundamental to the rationale of the research) is highlighted. Additionally, I recommend addressing the significance of the parasite and disease in men as well.
- From the material and methods section: “The chromatograms for each sequence were evaluated with the Mesquite software using the Chromaseq pakage[19] that partially makes use of the Phred and Phrap algorithms for base calling, assigning quality values to each called base, and assembling contigs [20,21]” Can the authors please elaborate on this sentence? Does it mean each amplicon was sequenced multiple times and calculated a consensus sequence (contig)?
- Could the authors please confirm whether the sequences obtained in this study have already been released? I have searched for entries with the provided accession numbers but have not been able to find them.
- Lines 160-163: T. tenax and T. gallinae were included as controls? Can the authors please elaborate on this sentence? From my understanding, were included for comparative purposes (which I agree is correct), and the controls are the T. vaginalis Pfor sequences downloaded from Genbank.
- Lines 197-200. Can the authors please specify? They obtained 16 ITS sequences (line 205), and 26 for the Pfor gene (line 214), is that correct?
- From Figure 1 legend: “Two groups (type 1 and type 2) are potentially grouped inside two large circles.” Can the authors please elaborate on the word “potentially”? the yellow points belong to GenBank entries which are already classified as types 1 and 2, is that correct? Or is it just a hypothesis?
- I would recommend the authors clarify/justify the relevance of Figure 6, given it is a 3D structure prediction based on partial sequences. But more importantly, it might not be relevant to highlight a couple of mutations in a couple of sequences once the genetic diversity analyses suggested purifying selection already. Unless the authors consider it could be contradictory.
- The authors stated in the discussion section: “No correlation between the main clinical characteristics of the samples and the results of phylogenetic analysis for Pfor A were found”. I suggest the authors rephrase this sentence, they did not present any clinical characteristics of the samples in the results section, nor correlation analysis. Although it can be true, there is no evidence or results to support such a statement. The same for the last sentence (lines 429-431).
- The authors should avoid inappropriate self-citations, for example, reference 21. In that case, the authors must reference the original manuscript rather than a manuscript from the authors where the software was used. Please revise the document and remove those entries.
Minor revisions
- Please double-check all the references and suitability, for example:
Reference 4 is not the correct reference for the draft genome
Reference 37, doesn’t mention Pfor genes at all (but it was used to support the non-enzymatic activity of Pfor genes).
- Please revise the document for gene/protein nomenclatures and use the universal guideline, Pfor genes are mentioned using multiple formats: PFOR, pfor, Pfor A, etc.
Minor editing of the english language required
Author Response
The manuscript by Martinez-Hernandez et al has improved from the previous version, and I would recommend it for publication after major revisions. There are still some inconsistencies that deserve attention.
- First of all, from my previous comments I suggested the authors use the new genome sequence release (PRJNA885811) rather than the previously drafted version (PRJNA16084). My suggestion aims for the authors to provide a more precise gene number and sequence for Pfor, and does not intend, in any case, the authors to sequence the whole genome, or to make any additional/extraordinary financial efforts.
R: We appreciate the suggestions. Analyzes were performed again including the new genome sequence release. This has strengthened the work because details of the sequences such as the number of copies and the location in the genome of T. vaginalis are now known. Minimal changes were observed in the phylogenetic trees and slight modifications were observed in the network and the table 1. Now, in this new version of our manuscript, the new GenBank accession numbers are included in the methodology section and the corresponding changes can be observed directly in the figures.
- While the authors have incorporated the isolation source in the title (females), pertinent information has been deleted in the introduction section, where the relevance to women (fundamental to the rationale of the research) is highlighted. Additionally, I recommend addressing the significance of the parasite and disease in men as well.
R: Previous information on the importance of trichomoniasis in women that had been deleted has been reinstated and a paragraph on the significance of the parasite in the disease in men has also been included. (page 1-2, lines 45-51)
- From the material and methods section: “The chromatograms for each sequence were evaluated with the Mesquite software using the Chromaseq pakage[19] that partially makes use of the Phred and Phrap algorithms for base calling, assigning quality values to each called base, and assembling contigs [20,21]” Can the authors please elaborate on this sentence? Does it mean each amplicon was sequenced multiple times and calculated a consensus sequence (contig)?
R: To clarify this sentence we included the following text: “in most samples, amplification assays were performed at least twice, both forward and reverse, to achieve the quality and concentration required for sequencing” (page 3, lines 120-122, and later “The chromatograms for each sequence were evaluated with the Mesquite software using the Chromaseq package [22] that partially makes use of the Phred and Phrap algorithms for base calling, assigning quality values to each called base, and assembling contigs [23,24], because more than one sequence was obtained for most of the samples” (Page3, lines 128-131). Since the concentration or quality (integrity) of DNA in some samples as “limiting reagent” were relevant problems, because samples were previously used in other study (page 4, lines 180-183), we decided sequenced more than twice in most samples and each amplicon was sequenced for forward and reverse. Now, with the new genome release (PRJNA885811), we know that only one copies of pfor A exist in the genome. The above was agrees with not observe variation when carrying out the consensus sequence.
- Could the authors please confirm whether the sequences obtained in this study have already been released? I have searched for entries with the provided accession numbers but have not been able to find them.
R: We apologize; the sequences will be available in the GenBank as soon as this manuscript is published.
- Lines 160-163:T. tenax and T. gallinae were included as controls? Can the authors please elaborate on this sentence? From my understanding, were included for comparative purposes (which I agree is correct), and the controls are the T. vaginalis Pfor sequences downloaded from Genbank.
R: Your assessment is correct; to clarify this sentence, the text was changed; now it reads: “Sequences of the ITS1+5.8S+ITS2 region for BA reconstructions, were MK172845-7, KM095107; KX459478-79,86; KX977518-20 and KC215389-90 from T. gallinae, T. tenax, T. brixi and T. stableri, respectively, using to T. stableri as outgroup; while for pfor A tree, sequences of T. tenax and T. gallinae (OCTD01001480 and MRSU01008911, respectively) were included as outgroup. In addition, some T. vaginalis sequences were downloaded from GenBank as controls.
- Lines 197-200. Can the authors please specify? They obtained 16 ITS sequences (line 205), and 26 for the Pfor gene (line 214), is that correct?
R: It is correct. The ITS1+5.8S+ITS2 was originally used as a tool for the molecular identification of T. vaginalis in samples with unclear microscopic observations, after that, the samples were used for the amplification and sequencing of pfor A. The following information is shown in the text: “Unfortunately, since the DNA obtained from the samples was used for a previous study [17], it was not possible to obtain amplicons of ITS1+5.8S+ITS2 and PFOR of sufficient quality and concentration for sequencing in all cases” (page 4, lines 190-193).
- From Figure 1 legend: “Two groups (type 1 and type 2) are potentially grouped inside two large circles.” Can the authors please elaborate on the word “potentially”? the yellow points belong to GenBank entries which are already classified as types 1 and 2, is that correct? Or is it just a hypothesis?
R: We appreciate your observation. We apologize for the mistake of using an ambiguous word in the figure caption. Now it can be read in the figure legend: “Two subgroups (1 and 2, respectively) can be differentiated, one close to the species T. gallinae and another to T. tenax; both subgroups are shown by two large circles”.
- I would recommend the authors clarify/justify the relevance of Figure 6, given it is a 3D structure prediction based on partial sequences. But more importantly, it might not be relevant to highlight a couple of mutations in a couple of sequences once the genetic diversity analyses suggested purifying selection already. Unless the authors consider it could be contradictory.
R: Figure 5 (in the previous version it was figure 6) shows the three-dimensional structure of the PFOR fragment from the pcr-sequencing assays that were performed in the present study. Certainly, only two mutations located far from cysteine-rich sites were found and in appearance they may not have an impact on the protein function, however, however, due to the experimental design and the results obtained in the present study, we do not have evidence that allows us to accept or rule out the real impact of these mutations on the function of the protein. Furthermore, as support for the figure, during the discussion we show examples of unique mutations in nitroreductases, which can confer resistance to metronizadol in T. vaginalis. (Page 11, lines 353-361).
Finally, there is information in the literature about single mutations in other ferredoxins (far from cysteine-rich sites), where their function in the ability to interact with other molecules or changes in their redox potential is affected (Holden HM. et al 1994, Zanetti G. et al 2011, Hurley JK. Et al 1994, Hurley JK. et al 1993). For these reasons, we consider it appropriate to show them since the information in the literature of PFOR mutations in parasites is scarce.
- The authors stated in the discussion section: “No correlation between the main clinical characteristics of the samples and the results of phylogenetic analysis for Pfor A were found”. I suggest the authors rephrase this sentence, they did not present any clinical characteristics of the samples in the results section, nor correlation analysis. Although it can be true, there is no evidence or results to support such a statement. The same for the last sentence (lines 429-431).
R: Thank you for your comment. Now, in this new corrected version of our manuscript, we included a supplementary table (Appendix A), in which association analysis between main clinical status and subclustering for pfor A is shown.
- The authors should avoid inappropriate self-citations, for example, reference 21. In that case, the authors must reference the original manuscript rather than a manuscript from the authors where the software was used. Please revise the document and remove those entries.
- Ok, references were checked and those inappropriate self-citations were changed by the original manuscript.
Minor revisions
- Please double-check all the references and suitability, for example:
Reference 4 is not the correct reference for the draft genome
Reference 37, doesn’t mention Pfor genes at all (but it was used to support the non-enzymatic activity of Pfor genes).
R: Thank you. The references were now carefully checked.
- Please revise the document for gene/protein nomenclatures and use the universal guideline, Pfor genes are mentioned using multiple formats: PFOR, pfor, Pfor A, etc.
R: Thank you again. According to https://www.biosciencewriters.com/Guidelines-for-Formatting-Gene-and-Protein-Names, Parasites/Worms: Gene symbols are italicized and generally composed of three to four letters, a hyphen, and an Arabic number (e.g., abu-1). Protein symbols are not italicized, and all letters are in upper-case (e.g., ABU-1); then, we used pfor A for gene and PFOR A for protein.
Round 2
Reviewer 3 Report (Previous Reviewer 3)
Major issues and concerns have been solved, and I suggest its acceptance for publication.
I suggest a final check for typos.
This manuscript is a resubmission of an earlier submission. The following is a list of the peer review reports and author responses from that submission.
Round 1
Reviewer 1 Report
The manuscript entitled "Polymorphisms of the internal transcribed spacer (ITS) and pyruvate:ferredoxin oxidoreductase (pfor) A gene of Trichomonas vaginalis recovered from clinical isolates" Title, abstract and overall rationale of work is written unsatisfactory. There are major concerns, which needs to be addressed before publication.
1) The authors provide a Polymorphisms of the internal transcribed spacer (ITS) and pyruvate:ferredoxin oxidoreductase (pfor) A gene of Trichomonas vaginalis recovered from clinical isolates. The report is clear and straightforward. Eventual acceptance should not be an issue, however further controls are necessary. Furthermore, numerous errors in content are found throughout the manuscript that needs to be corrected. In the title part: No need to use abbreviation and I recommend to remove this.
2) The introduction part is written well but little bit lengthy and I suggest the author to reduce little bit.
3) Material method section is well describe and details information are there,
4) Author need to increase resolution of all figures specially figure 2 and 5.
5) Result and Discussion section: This section author need to improve because author written the results part but they do not discuss properly and I saw the lack of discussion in this manuscript. I recommend author, they should elaborate the discussion part and author need to revise and compare the study with relevant study.
6) How many times this experiment repeated, kindly provide the details.
7) Author need to incorporate conclusion section author must be write the limitation of this study and significance. Moreover, author also need to write future prospective of this study.
8) Some references are too long and author need to revise for example reference no 26, 27, 30 and other. I suggest author to revise if other latest manuscript is available in the same information.
Author Response
1) The authors provide a Polymorphisms of the internal transcribed spacer (ITS) and pyruvate:ferredoxin oxidoreductase (pfor) A gene of Trichomonas vaginalis recovered from clinical isolates. The report is clear and straightforward. Eventual acceptance should not be an issue; however further controls are necessary. Furthermore, numerous errors in content are found throughout the manuscript that needs to be corrected. In the title part: No need to use abbreviation and I recommend to remove this.
R: Firstly, on behalf of all the co-authors, we thank you for your comments and suggestions, which have allowed us to improve our manuscript. Below, we address all comments point by point, clarifying the subsequent modifications. Heeding to reviewer 2, title was changed and abbreviations were deleted.
2) The introduction part is written well but little bit lengthy and I suggest the author to reduce little bit.
R: Ok, the introduction was edited and reduced, now it focuses on PFOR and ITS markers
3) Material method section is well describing and details information are there.
R: Following the recommendations of the reviewer 3, some processes have been detailed.
4) Author need to increase resolution of all figures specially figure 2 and 5.
R: Resolutions of all figures were increased
5)Result and Discussion section: This section author need to improve because author written the results part but they do not discuss properly and I saw the lack of discussion in this manuscript. I recommend author, they should elaborate the discussion part and author need to revise and compare the study with relevant study.
R: Results and discussion are in two separate sections. We have extended the discussion of the results (Page 11, lines 316-322; page 12, lines 326-331; 336-338; page 13, lines 379-388).
6) How many times this experiment repeated, kindly provide the details.
R: Following text was added: “DNA extractions and PCRs were performed for each sample from two to four times to reach the concentration and quality recommended for sequencing” (Page 4, lines 168-170)
7) Author need to incorporate conclusion section author must be write the limitation of this study and significance. Moreover, author also need to write future prospective of this study.
R: Ok, the following text was added: “Since, the present study was based on sequence analysis of a fragment of the pfor A gene and ITS marker from clinical isolates of T. vaginalis recovered from 26 patients, the small sample size constitutes a potential bias in the analysis of their genetic variability; however, in both markers it was possible to yield robust phylogenetic inferences that allows identifying among different clusters and haplotypes, as well as identify a probably expansion population process in this parasite. In addition, the development of future studies using next-generation sequencing (NGS) for a more extensive search and accurate results applied to a case-control design, would help to clarify the etiological role of these markers in clinical isolates of T. vaginalis, as well as, to include the study of samples from other populations and male patients.
- Conclusion
Our results suggest that the role of genetic variability of PFOR and ITS could be not significant in the symptomatology of this pathogen; however, their utility as genus and species markers in trichomonads is promising”. (Page 13, lines 408-410).
8) Some references are too long and author need to revise for example reference no 26, 27, 30 and other. I suggest author to revise if other latest manuscript is available in the same information.
R: References were checked throughout the text; in this new version of our manuscript, reference is made to the most important or recent articles with the same information.
Reviewer 2 Report
This study is interesting and gathers a great amount of evidence regarding T. vaginalis polymorphism and the potential impact on its pathogenicity. The authors have presented their findings effectively and have successfully discussed them. However, while reading through it, I encountered a few concerns, which could be easily addressed through minor revisions. Specifically, the following topics need to be better addressed:
Title – It should be more informative and highlight the study's main findings; Considering that the investigation was performed in clinical samples obtained from female patients, it should be informed in the title.
Abstract – This section is adequate, but the objective could be improved by citing the evaluation of possible implications in the host-parasite relationship.
Introduction – It provides a suitable background of the subject. Is it the first study that examines such polymorphisms? In the case of an affirmative answer, it must be highlighted.
Material and methods – This section is adequate, well-written, and organized.
The authors discuss part of their results in this section, which is inadequate because the results and discussion sections are separate. I suggest the authors combine these sections instead of keeping them in the current version. A combination of these sections would improve readability and comprehension.
Regarding the limitation of the samples, what are the implications of the conclusions? Is it any potential flaw in the data interpretation? Authors should discuss it.
Is it possible to suggest that the results obtained from the samples of infected females would be expected for male-infected patients? I wondered if the lower infectivity of T. vaginalis in men is related to this polymorphism because it affects the pathogenicity. It should be briefly discussed.
Lastly, the authors must include a brief paragraph approaching the potential limitations of their study and clarifying the concrete outcomes of their findings in the field. It must include perspectives for future studies.
Author Response
This study is interesting and gathers a great amount of evidence regarding T. vaginalis polymorphism and the potential impact on its pathogenicity. The authors have presented their findings effectively and have successfully discussed them. However, while reading through it, I encountered a few concerns, which could be easily addressed through minor revisions. Specifically, the following topics need to be better addressed:
R: Dear review, thank you very much for your comments, since the changes made have allowed us to improve and detail the information in our manuscript.
Title – It should be more informative and highlight the study's main findings; Considering that the investigation was performed in clinical samples obtained from female patients, it should be informed in the title.
R: Title was changed; now, it reads: Genetic variability of the internal transcribed spacer and pyruvate:ferredoxin oxidoreductase partial gene of Trichomonas vaginalis from female patients
Abstract – This section is adequate, but the objective could be improved by citing the evaluation of possible implications in the host-parasite relationship.
R: The objective was supplemented regarding the evaluation of possible implications in the host-parasite relationship (Page 1, lines 26-27)
Introduction – It provides a suitable background of the subject. Is it the first study that examines such polymorphisms? In the case of an affirmative answer, it must be highlighted.
R: The following text was added: “It is important to highlight that, to date, PFOR studies have been focused on their characterization and function depending on their localization, but whether pfor genes present genetic variations and are associated with pathogenicity or the development of specific symptoms have not been described” (Page 2, lines 66-70) and “Although there are some reports about the usefulness of ITS1-5.8S rRNA-ITS2 as a marker for differentiation among T. vaginalis isolates, information obtained using this locus is still scarce [13-16]”. (Page 2, lines 77-79)
Material and methods – This section is adequate, well-written, and organized.
R: Following the recommendations of the reviewer 3, some processes were detailed.
Result and Discussion section- The authors discuss part of their results in this section, which is inadequate because the results and discussion sections are separate. I suggest the authors combine these sections instead of keeping them in the current version. A combination of these sections would improve readability and comprehension.
R: Thank you, we are agreeing with you; but, since the Microorganisms Original Articles Format requires that all manuscripts must contain Results and Discussion as separated sections (https://www.mdpi.com/journal/microorganisms/instructions); so, we cannot combine both sections. However, we revised both section, and in this new version of our manuscript, those paragraphs of discussion in the results section were moved to the discussion section.
Regarding the limitation of the samples, what are the implications of the conclusions? Is it any potential flaw in the data interpretation? Authors should discuss it.
R: Coincidentally, reviewer 1 made a similar suggestion; so, the following text was added: “Since, the present study was based on sequence analysis of a fragment of the pfor A gene and ITS marker from clinical isolates of T. vaginalis recovered from 26 patients, the small sample size constitutes a potential bias in the analysis of their genetic variability; however, in both markers it was possible to yield robust phylogenetic inferences that allows identifying among different clusters and haplotypes, as well as identify a probably expansion population process in this parasite. In addition, the development of future studies using next-generation sequencing (NGS) for a more extensive search and accurate results applied to a case-control design, would help to clarify the etiological role of these markers in clinical isolates of T. vaginalis, as well as, to include the study of samples from other populations and male patients.” (Page 13, lines 387-397).
Is it possible to suggest that the results obtained from the samples of infected females would be expected for male-infected patients? I wondered if the lower infectivity of T. vaginalis in men is related to this polymorphism because it affects the pathogenicity. It should be briefly discussed.
R: Thank you, this is a very relevant question; however, due to the nature of the present study and reviewer 3's recommendations, we apologize but we chose to omit discussing this aspect to avoid " a too verbose discussion" and focus only on our results. Nevertheless, this concern is shown in this new version of the manuscript: “…the development of future studies using next-generation sequencing (NGS) for a more extensive search and accurate results applied to a case-control design, would help to clarify the etiological role of these markers in clinical isolates of T. vaginalis, as well as, to include the study of samples from other populations and male patients. (Page 13, lines 393-397)
Lastly, the authors must include a brief paragraph approaching the potential limitations of their study and clarifying the concrete outcomes of their findings in the field. It must include perspectives for future studies.
R: Ok, you are right, and in concordance with those previous concerns the following text was added: “Since, the present study was based on sequence analysis of a fragment of the pfor A gene and ITS marker from clinical isolates of T. vaginalis recovered from 26 patients, the small sample size constitutes a potential bias in the analysis of their genetic variability; however, in both markers it was possible to yield robust phylogenetic inferences that allows identifying among different clusters and haplotypes, as well as identify a probably expansion population process in this parasite. In addition, the development of future studies using next-generation sequencing (NGS) for a more extensive search and accurate results applied to a case-control design, would help to clarify the etiological role of these markers in clinical isolates of T. vaginalis, as well as, to include the study of samples from other populations and male patients. (Page 13, lines 387-397).
Reviewer 3 Report
In this manuscript, the authors amplified, sequenced, and analyzed the polymorphisms of the ITS region and the pfor gene of T. vaginalis. Given the divergence the authors found, they hypothesized this gene can be used as a molecular marker for this parasite.
Unfortunately, I consider the manuscript unsuitable for publication for many reasons.
- Sequencing methods/technology and results are unclear.
- Sequence alignments are biased.
- The authors should use the new release genome sequence for a more extensive search and accurate results.
- All the figures are redundant, barely discussed, or not discussed at all.
- Conclusions and discussions are not supported by the results.
N/A
Author Response
In this manuscript, the authors amplified, sequenced, and analyzed the polymorphisms of the ITS region and the pfor gene of T. vaginalis. Given the divergence the authors found, they hypothesized this gene can be used as a molecular marker for this parasite.
Unfortunately, I consider the manuscript unsuitable for publication for many reasons.
R: We understood that several aspects of our manuscript needed to be clarified; so, we carried out an extensive revision, as well as core modifications to our manuscript in accordance with the recommendations of all the reviewers; now, in this new corrected version, our manuscript has been improved.
- Sequencing methods/technology and results are unclear.
R: To clarify this process, the following texts were added: “To analyze the pfor gene, specific primers for variant A of different T. vaginalis sequences were designed at the laboratory based on information available in GenBank (se-quences AY661465, U16822, and XM_001582310). A suitable region of ~852 bp was chosen for amplification by the forward primer 5´ TCAAGGTCCACCTCTTCCGCCC 3´ and the reverse primer 5ʹ GCATGGTCGACCATGTCCCAG 3´. This region includes the important sites of the enzyme, such as the domain II and III of the ferredoxin oxidoreductase site, which are all located between amino acids 305 and 603 of the protein” (Page 3, lines 100-106) and “the amplicons were purified using an AxyPrep PCR clean-up kit (Axigen Biosciences, Union City, CA, USA) and sequenced on both strands in an ABI prism sequencer (Applied Biosystems, Foster City, CA, USA) using the Sanger method and chromatograms were obtained.” (Page 3, lines 130-134)
- Sequence alignments are biased
R: To avoid misunderstandings to readers, following texts was added: “The chromatograms for each sequence were evaluated with the Mesquite software using the Chromaseq package [19] that partially makes use of the Phred and Phrap algorithms for base calling, assigning quality values to each called base, and assembling contigs [20,21].” (Page 3, lines 120-123) and “multiple alignments were performed using CLUSTAL W and MUSCLE programs with manual adjustment in the MEGA program [22].” (Page 3, lines 225.227).
In addition, enclosed please find two files that shown the final alignments.
- The authors should use the new release genome sequence for a more extensive search and accurate results.
R: You are certainly right; however, as you know, many research sites do not have the financial resources or equipment to carry out full experiments. However, partial and interesting findings of medical-biological interest, as in the present study, are worth publishing.
Heeding to your concern, the following text has been added: “Since, the present study was based on sequence analysis of a fragment of the pfor A gene and ITS marker from clinical isolates of T. vaginalis recovered from 26 patients, the small sample size constitutes a potential bias in the analysis of their genetic variability; however, in both markers it was possible to yield robust phylogenetic inferences that allows identifying among different clusters and haplotypes, as well as identify a probably expansion population process in this parasite. In addition, the development of future studies using next-generation sequencing (NGS) for a more extensive search and accurate results applied to a case-control design, would help to clarify the etiological role of these markers in clinical isolates of T. vaginalis, as well as, to include the study of samples from other populations and male patients.”. (Page 13, lines 387-397)
- All the figures are redundant, barely discussed, or not discussed at all.
R: In this new version of our manuscript, the discussion of the figures was expanded (Page 11, lines 308-311, page 12, lines 334-339, page 12-13, lines 370-380). To avoid redundancy in the figures, figure 5 was moved to appendix A
- Conclusions and discussions are not supported by the results.
R: As you can see, the Discussion section was improved focused on the results obtained and the conclusion was rewritten (page 13, lines 408-410).